# Auto-Encoding Knowledge Graph for Unsupervised Medical Report Generation

**Fenglin Liu[1,2], Chenyu You[4], Xian Wu[5], Shen Ge[5], Sheng Wang[3],*Xu Sun[1]***

[1]MOE Key Laboratory of Computational Linguistics, School of EECS, Peking University
[2]School of ECE, Peking University
[3]Paul G. Allen School of Computer Science and Engineering, University of Washington
[4]Department of Electrical Engineering, Yale University    [5]Tencent
`{fenglinliu98, xusun}@pku.edu.cn, chenyu.you@yale.edu`
`{kevinxwu, shenge}@tencent.com, swang@cs.washington.edu`

## Abstract

Medical report generation, which aims to automatically generate a long and coherent report of a given medical image, has been receiving growing research interests. Existing approaches mainly adopt a supervised manner and heavily rely on coupled image-report pairs. However, in the medical domain, building a large-scale image-report paired dataset is both time-consuming and expensive. To relax the dependency on paired data, we propose an unsupervised model Knowledge Graph Auto-Encoder (KGAE) which accepts independent sets of images and reports in training. KGAE consists of a pre-constructed knowledge graph, a knowledge-driven encoder and a knowledge-driven decoder. The knowledge graph works as the shared latent space to bridge the visual and textual domains; The knowledge-driven encoder projects medical images and reports to the corresponding coordinates in this latent space and the knowledge-driven decoder generates a medical report given a coordinate in this space. Since the knowledge-driven encoder and decoder can be trained with independent sets of images and reports, KGAE is unsupervised. The experiments show that the unsupervised KGAE generates desirable medical reports without using any image-report training pairs. Moreover, KGAE can also work in both semi-supervised and supervised settings, and accept paired images and reports in training. By further fine-tuning with image-report pairs, KGAE consistently outperforms the current state-of-the-art models on two datasets.

## 1 Introduction

Medical images, such as radiology and pathology images, and their corresponding reports are widely used for clinical diagnosis and treatment [8, 11]. A medical report is usually a paragraph of multiple sentences which describes both the normal and abnormal findings in the medical image. In clinical practice, writing a report can be time-consuming and tedious for experienced radiologists, and error-prone for inexperienced radiologists [4]. Therefore, given the large volume of medical images, automatically generating reports can improve current clinical practice in diagnostic radiology and assist radiologists in clinical decision-making [15, 25]. Specifically, it can relieve radiologists from such heavy workload and alert radiologists of the abnormalities to avoid misdiagnosis and missed diagnosis. Therefore, automatic medical report generation attracts remarkable attention in both artificial intelligence and clinical medicine.

Recently, inspired by the great success of neural machine translation [2, 38, 49, 50, 48], image captioning [44, 39, 46, 28, 30, 29] and medical imaging analysis [47, 51, 52], the data-driven deep

---

*Corresponding authors.

35th Conference on Neural Information Processing Systems (NeurIPS 2021).

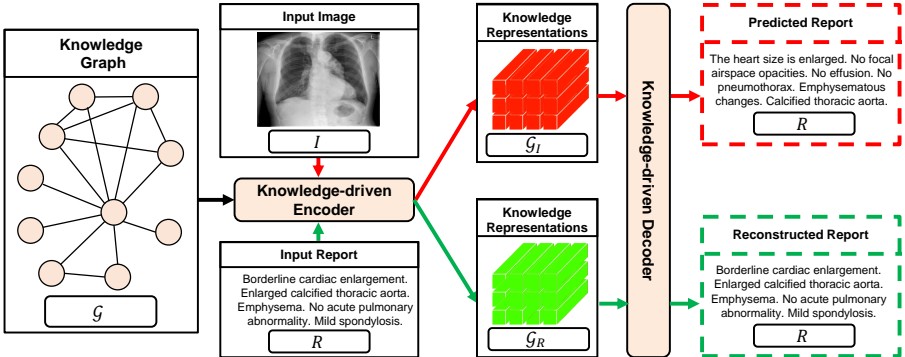

Figure 1: Illustration of our Knowledge Graph Auto-Encoder, which consists of a pre-constructed knowledge graph, a knowledge-driven encoder and a knowledge-driven decoder. The Green and Red lines denote the data flow in the training process and testing process of report generation, respectively.

neural models, particularly those based on the encoder-decoder frameworks [15, 16, 25, 22, 41, 45, 53, 54, 6], have achieved great success in advancing the state-of-the-art of medical report generation. However, these models are trained in a supervised learning manner and heavily rely on labeled paired image-report datasets [9, 17], which are not easy to acquire in the real world. Specifically, the medical-related data can only be manually labeled by professional radiologists, and also involves privacy issues. Therefore, the medical report generation datasets are particularly labor-intensive and expensive to obtain. As a result, the scales of existing widely-used datasets for medical report generation models [16, 25, 22, 23], i.e., MIMIC-CXR (0.22M samples) [17] and IU X-ray (4K samples) [9], are relatively small compared to image recognition datasets, e.g., ImageNet (14M samples) [10], and image captioning datasets, e.g., Conceptual Captions (3.3M samples) [36]. In addition, the MIMIC-CXR and IU X-Ray datasets only include Chest X-Ray images, for other types of medical images (MRI, Dermoscopy, Retinal, etc.) of other body parts (brain, skin, eye, etc.), the image-report pairs could be much less or even unavailable. Therefore, to relax the reliance on the paired data sets, making use of all available data, like independent image or report sets, is becoming increasingly important.

In this paper, we propose an unsupervised model Knowledge Graph Auto-Encoder (KGAE), which utilizes independent sets of images and reports in training (the image and report set are separate and have no overlap). KGAE consists of a pre-constructed knowledge graph, a knowledge-driven encoder and a knowledge-driven decoder. As shown in Figure 1, the knowledge graph works as the shared latent space of images and reports. The knowledge-driven encoder can take either the image $I$ or the report $R$ as queries and project them to corresponding coordinates $\mathcal{G}_I$ and $\mathcal{G}_R$ in the latent space. In this manner, since $\mathcal{G}_I$ and $\mathcal{G}_R$ share the same latent space, we can use positions in latent space to measure the relationship between images and reports which narrows the gap between visual and textual domains. In brief, to bridge the gap between vision and language domains without training on the pairs of images and reports, we adopt the knowledge graph to create a latent space and propose a knowledge-driven encoder, which includes a common mapping function to project images and reports to the same latent space. As a result, our encoder can extract the image and report knowledge representations, i.e., the knowledge related to the image and report, they (image, report knowledge) share the common latent space, which allows our model to bridge the gap between vision and language domains without the training on the pairs of image and report. Next, we introduce the knowledge-driven decoder to exploit $\mathcal{G}_I$ and $\mathcal{G}_R$ to generate the report. In the training stage, we estimate the parameters of the decoder by reconstructing the input report $R$ based on $\mathcal{G}_R$, i.e., $R \rightarrow \mathcal{G}_R \rightarrow R$ auto-encoding pipeline; In the prediction stage, we directly input $\mathcal{G}_I$ into the trained decoder to generate the report. In this way, our approach can produce desirable reports without any labeled image-report pairs.

Overall, the contributions of this paper are as follows:

- In this paper, we make the first attempt to conduct unsupervised medical report generation where the image-report pairs are not available. To this end, we propose the Knowledge Graph Auto-Encoder (KGAE). By leveraging a pre-constructed knowledge graph, we introduce the knowledge-driven encoder and decoder which are trained with independent sets of

images and reports. According to the experimental results, the unsupervised KGAE can even outperform several supervised approaches.

- In addition to the unsupervised mode, KGAE can also be applied in a semi-supervised or supervised manner. Under the semi-supervised setting, by using only 60% of paired dataset, KGAE is able to achieve competitive results with current state-of-art models; Under the supervised setting, by training on fully paired datasets as in existing works, KGAE can set new state-of-the-art performances on the IU X-ray and MIMIC-CXR, respectively.

- The analysis, both quantitative and qualitative, as well as a human evaluation conducted by professional radiologists, further proves the effectiveness of our approach.

## 2   Related Works

Medical report generation aims to generate a relatively long paragraph to describe a given medical image. It is similar to the image captioning task [5], which aims to generate a sentence to describe a given image. In image captioning, the encoder-decoder framework [39], where the encoder [20, 12] computes visual representations for the image and the decoder [13, 38] generates a target sentence based on the visual representations, has achieved great success [1, 31, 35, 44]. However, instead of only generating one single sentence, medical report generation aims to generate a long paragraph including multiple structured sentences that describe both the normal and abnormal parts [25, 15]. To this end, given the success of encoder-decoder framework on image captioning, most existing medical report generation models attempt to exploit the hierarchical LSTM (HLSTM) [15, 16, 19] or Transformer [6] to generate an accurate, long and coherent report. However, existing models require the paired image-report datasets, which are time-consuming and expensive to collect. In this paper, we propose the unsupervised model Knowledge Graph Auto-Encoder (KGAE) which doesn't need paired images and reports.

Note that although the knowledge graph has been integrated in existing medical report generation models [54, 22], these approaches are supervised and require paired images and reports. Thus, their objectives and motivations of using knowledge graph are different from our work. In detail, existing knowledge-graph based medical report generation methods aim to adopt the knowledge graph to boost the performance of supervised models. However, in our work, we aim to generate a medical report without using any coupled image-report training pairs, i.e., unsupervised medical report generation. A key challenge of unsupervised medical report generation is to bridge the gap between vision and language domains. To this end, we adopt the knowledge graph to create a latent space and propose a knowledge-driven encoder to project image and report to the same latent space.

## 3   Approach

We first formulate the conventional supervised medical report generation problems; Then, we describe the proposed Knowledge Graph Auto-Encoder for unsupervised medical report generation in detail.

### 3.1   Conventional Supervised Medical Report Generation Models

Given a medical image $I$, the goal is to generate a descriptive report $R$. Most models [15, 16, 6] normally include an image encoder and a report decoder, which can be formulated as:

$$\text{Image Encoder} : I \rightarrow I'; \ \text{Report Decoder} : I' \rightarrow R, \tag{1}$$

where $I' \in \mathbb{R}^{N_I \times d}$ denotes the image embeddings extracted by the image encoder, e.g., ResNet-50 [12]. Then, $I'$ is used to guide the generation of the target report $R$ in the report decoder, e.g., LSTM [13] and Transformer [38]. During training, given the ground truth report for the input image, we can train the encoder-decoder model by minimizing a supervised training loss, e.g., cross-entropy loss. However, paired image-report ($I$-$R$) datasets are particularly labor-intensive and expensive to obtain. Therefore, in this paper, we aim to relax the reliance on the paired datasets and make use of the independent sets of images and reports instead.

### 3.2   Knowledge Graph Auto-Encoder

As shown in Figure 1, the proposed KGAE includes a knowledge graph, a knowledge-driven encoder and a knowledge-driven decoder, which will be described in detail in the following sections.

**Knowledge Graph**    The motivation of using the knowledge graph is that writing medical reports usually requires particular domain knowledge [11, 22]. Therefore, we can utilize the knowledge

graph, which models the domain-specific knowledge structure, to serve as a bridge to correlate the knowledge representations of the visual and textual domains. In particular, we construct an off-the-shelf global medical knowledge graph $\mathcal{G} = (V, E)$ covering the common abnormalities and normalities, where $V = \{v_i\}_{i=1:N_{\text{KG}}} \in \mathbb{R}^{N_{\text{KG}} \times d}$ is a set of nodes and $E = \{e_{i,j}\}_{i,j=1:N_{\text{KG}}}$ is a set of edges. In detail, based on the report corpus, i.e., MIMIC-CXR [17], we consider the $N_{\text{KG}}$ frequent clinical abnormalities (e.g., "enlarged heart size", "pleural effusion", and "bibasilar consolidation") and normalities (e.g., "heart size is normal" and "lungs are clear") as nodes. The edge weights are calculated by the normalized co-occurrence of different nodes computed from report corpus. Please note that this knowledge graph is built purely from the training set of MIMIC-CXR, thus there is no label leakage. After that, the knowledge graph is embedded by a graph embedding module, i.e., graph convolution network [32, 24], which is implemented as follows:

$$v'_i = v_i + \text{ReLU}(\sum_{j=1}^{N_{\text{KG}}} e_{i,j} W_v v_j), \tag{2}$$

where $\text{ReLU}(\cdot)$ represents the ReLU activation function and $W_v \in \mathbb{R}^{d \times d}$ is the learnable matrix. As a result, we can acquire a set of node embeddings $V' = \{v'_1, v'_2, \ldots, v'_{N_{\text{KG}}}\} \in \mathbb{R}^{N_{\text{KG}} \times d}$. It is worth noting that more complex graph structures could be constructed by using more large-scale medical textbooks. Therefore, our model is not limited to the currently constructed graph and could provide a good basis for the research of unsupervised medical report generation for other domains.

**Knowledge-driven Encoder**    The knowledge-driven encoder (KE) is designed to utilize the knowledge graph $\mathcal{G}$ to extract knowledge representations ($\mathcal{G}_I \in \mathbb{R}^{N_I \times d}$ and $\mathcal{G}_R \in \mathbb{R}^{N_R \times d}$) of both image $I$ and report $R$ to bridge the vision and the language domains, which can be formulated as:

$$\mathcal{G}_I = \text{KE}_I(I, \mathcal{G}); \;\; \mathcal{G}_R = \text{KE}_R(R, \mathcal{G}). \tag{3}$$

In implementations, as shown in Figure 1, we first adopt the ResNet-50 [12] and the Transformer [38] as the image embedding module and report embedding module to embed the image $I$ and report $R$, acquiring the image embeddings $I' \in \mathbb{R}^{N_I \times d}$ and report embeddings $R' \in \mathbb{R}^{N_R \times d}$, respectively. Next, to extract the knowledge representations from knowledge graph $\mathcal{G}$, we adopt the attention mechanism [38] to implement the KE. The motivation stems from that the attention mechanism can compute the association weights between different features and allows probabilistic many-to-many relations instead of monotonic relations, as in [38, 44, 29]. As a result, we take $I'$ and $R'$ as the queries, and take the knowledge graph $\mathcal{G}$ (i.e., $V' = \{v'_1, v'_2, \ldots, v'_{N_{\text{KG}}}\} \in \mathbb{R}^{N_{\text{KG}} \times d}$) as the lookup matrix:

$$\mathcal{G}_I = \text{KE}_I(I, \mathcal{G}) = \mathcal{F}(\text{Attention}_I(I', V')); \;\; \mathcal{G}_R = \text{KE}_R(R, \mathcal{G}) = \mathcal{F}(\text{Attention}_R(R', V'))$$

$$\text{Attention}(x, y) = \text{softmax}\left(\frac{x W_{\text{q}} (y W_{\text{k}})^\top}{\sqrt{d}}\right) y W_{\text{v}}. \tag{4}$$

where $x \in \mathbb{R}^{N_x \times d_x}, y \in \mathbb{R}^{N_y \times d_y}$; $W_{\text{q}} \in \mathbb{R}^{d_x \times d_x}$, $W_{\text{k}} \in \mathbb{R}^{d_y \times d_y}$ and $W_{\text{v}} \in \mathbb{R}^{d_y \times d_y}$ ($d_x = d_y$) are learnable parameters. $\mathcal{F}$ is implemented as two fully-connected (FC) layers with a ReLU in between, i.e., FC-ReLU-FC. The resulted $\mathcal{G}_I \in \mathbb{R}^{N_I \times d}$ and $\mathcal{G}_R \in \mathbb{R}^{N_R \times d}$ turn out to be a set of attended (i.e., extracted) knowledge related to the image and report. The knowledge representations $\mathcal{G}_I$ and $\mathcal{G}_R$ share the common latent space which bridges the vision and the language domains. Note that the $\mathcal{F}$ used in $\text{KE}_I$ and $\text{KE}_R$ shares the same parameters and is thus introduced to further boost the bridging capabilities of different modalities. It means that we adopt two KEs, i.e., $\text{KE}_I$ and $\text{KE}_R$, and each KE includes an attention model as well as a **common mapping function** $\mathcal{F}$. The only shared weights of $\text{KE}_I$ and $\text{KE}_R$ are the parameters of $\mathcal{F}$, and the parameters of attention models are independent.

**Knowledge-driven Decoder**    The decoder is designed to generate the reports based on the graph representations $\mathcal{G}_I$ or $\mathcal{G}_R$. For clarity, we use $\mathcal{G}_k$ to represent the $\mathcal{G}_R$ and $\mathcal{G}_I$ during the training and testing stages, respectively. In implementations, since medical report generation requires generating a long paragraph, we choose the (three-layer) Transformer [38] as the basic module of our decoder and incorporate the proposed Knowledge-driven Attention (KA) to effectively model the long sequences.

At each decoding step $t$, the decoder takes the embedding of current input word $x_t = w_t + e_t \in \mathbb{R}^{2d}$ as input, where $w_t$ and $e_t$ denote the word embedding and fixed position embedding, respectively, and generate each word $r_t$ in report $R = \{r_1, r_2, \ldots, r_T\}$, which can be defined as follows:

$$\begin{aligned} h_t &= \text{Attention}(x_t, x_{1:t}) \\ h'_t &= \text{KA}(h_t, \mathcal{G}_k, B) \end{aligned}, \text{ where } \mathcal{G}_k = \begin{cases} \mathcal{G}_R, & \text{Training} \\ \mathcal{G}_I, & \text{Testing} \end{cases}; \; r_t \sim p_t = \text{softmax}(\text{FFN}(h'_t) W_p), \tag{5}$$

where FFN stands for Feed-Forward Network in the original Transformer [38]; $W_p \in \mathbb{R}^{2d \times |D|}$ is the learnable parameter ($|D|$ denotes the vocabulary size); $B$ represents the knowledge bank in the KA.

In particular, the knowledge-driven attention (KA) is inspired by the success of many works that attempt to augment a working memory into the network, e.g., memory network. Memory network preserves a dynamic knowledge base for subsequent inference [37, 21, 42, 43, 40, 46, 33]. For example, [37, 21] and [42, 43] proposed to preserve the context of a document and visual knowledge to solve language modeling and visual question answering, respectively. In this paper, since the knowledge representations $\mathcal{G}_k$ are extracted from the off-the-shelf global knowledge graph $\mathcal{G}$, we incorporate a knowledge memory mechanism to distill and preserve the fine-grained medical knowledge related to report generation task, which encourages our decoder to generate more accurate and desirable reports. Specifically, we introduce the knowledge bank $B = \{b_1, b_2, \ldots, b_{N_B}\} \in \mathbb{R}^{N_B \times d}$, where $N_B$ stands for the total number of the knowledge corresponding to report generation. As a result, given the graph representations $\mathcal{G}_k \in \mathbb{R}^{N_k \times d}$, the knowledge memory mechanism is defined as follows:

$$B_k = \text{softmax}\left(\mathcal{G}_k B^\top\right) B. \tag{6}$$

Through above operation, each feature/vector in the $\mathcal{G}_k \in \mathbb{R}^{N_k \times d}$ can distill the fine-grained knowledge preserved in $B$ to acquire $B_k \in \mathbb{R}^{N_k \times d}$ for accurate report generation.

Based on the above mechanism, the knowledge-driven attention in Eq. (5) is defined as:

$$h_t' = \text{KA}(h_t, \mathcal{G}_k, B) = \text{Attention}(h_t, [\mathcal{G}_k; B_k]) = \text{Attention}\left(h_t, \left[\mathcal{G}_k; \text{softmax}\left(\mathcal{G}_k B^\top\right) B\right]\right). \tag{7}$$

where $[\cdot; \cdot]$ denotes the concatenation operation. It is worth noting that these operations are all differentiable, thus the bank $B$ can be learned in an end-to-end fashion.

In our subsequent analysis, we will show that the introduced knowledge memory mechanism indeed distills and preserves the desired medical knowledge, and thus boost the generation of reports.

### 3.3 Implementation Details

**Unsupervised Training Details**   To train our KGAE in an unsupervised manner, instead of using the paired image-report dataset in the conventional supervised model, we only require an image set CheXpert [14], which includes 224,316 X-ray images, and a separate report corpus MIMIC-CXR [17] + IU X-ray [9], which includes 222,758 + 2,770 = 225,528 reports[2].

In detail, to train our knowledge-driven encoder $\text{KE}_I$ and $\text{KE}_R$ (see Eq. (4)), we feed the $\mathcal{G}_I$ and $\mathcal{G}_R$ into a common multi-label classification network [54, 15] trained with binary cross entropy loss for 14 common radiographic observations classification[3]. In this way, our encoder can extract the knowledge representations $\mathcal{G}_I$ and $\mathcal{G}_R$ of both image and report in a common latent space, effectively bridging the vision and the language domains. To train the knowledge-driven decoder, as well as the knowledge bank $B$ (see Eq. (5)), since there are no coupled image-report pairs, we propose to reconstruct the report $R$ based on the $\mathcal{G}_R$. Therefore, through Eq. (5), taking the input report $R = \{r_1, r_2, \ldots, r_T\}$ as the ground truth report, we can train our approach by minimizing the cross-entropy loss:

$$L_{\text{CE}} = -\sum_{t=1}^{T} \log\left(p\left(r_t \mid r_{1:t-1}\right)\right). \tag{8}$$

In this way, we can train our decoder in the $R \rightarrow \mathcal{G}_R \rightarrow R$ auto-encoding pipeline.

During testing, we first adopt the knowledge-driven encoder to extract the knowledge representations $\mathcal{G}_I$ of the test image (see Eq. (4)). Then, we directly feed $\mathcal{G}_I$ into the decoder to generate final report in the $I \rightarrow \mathcal{G}_I \rightarrow R$ pipeline (see Eq. (5)). In this way, our approach can relax the reliance on the image-report pairs. In our following experiments, we validate the effectiveness of our approach, which even outperforms some supervised approaches.

**Semi-Supervised and Supervised Training Details**   To further validate the effectiveness of our approach, we fine-tune the *unsupervised KGAE* using partial and full image-report pairs to acquire the *KGAE-Semi(-Supervised)* and *KGAE-Supervised*, respectively, where the former can evaluate the performance of our approach under limited labeled pairs for training and the latter can compare the

---

[2]There are no paired image-report samples between CheXpert and MIMIC-CXR+IU X-ray.

[3]Atelectasis, Cardiomegaly, Consolidation, Edema, Enlarged Cardiomediastinum, Fracture, Lung Lesion, Lung Opacity, No Finding, Pleural Effusion, Pleural Other, Pneumonia, Pneumothorax, Support Devices.

performance of KGAE with state-of-the-art supervised approaches. In the (semi-)supervised setting, given the image-report pairs, i.e., $I$-$R$, we first incorporate the original visual information into the knowledge representation $\mathcal{G}_I$, and then train our KGAE by generating the ground truth report in the $I \rightarrow \mathcal{G}_I \rightarrow R$ pipeline and minimizing the cross-entropy loss in Eq. (8). During testing, we also follow the unsupervised setting to generate the final report in the $I \rightarrow \mathcal{G}_I \rightarrow R$ pipeline.

## 4 Experiments

We first introduce the datasets, metrics and detailed settings used for evaluation. Then, we present the evaluation of our approach under the unsupervised, semi-supervised and supervised training settings.

### 4.1 Datasets, Metrics and Settings

**Datasets** In this paper, we adopt the test sets of IU X-ray [9] and MIMIC-CXR [17] for evaluation. All protected health information (e.g., patient name and date of birth) was de-identified. In particular, the IU X-ray [9] is a widely-used public benchmark dataset for medical report generation and contains 7,470 chest X-ray images associated with 3,955 fully de-identified medical reports. Each report is composed of impression, findings and indication sections, etc. [22]. Following [25], our method also focuses on the findings section as it is the most important component of reports. Then, following [16, 22, 25, 6], we randomly select 70%-10%-20% image-report pairs of dataset to form the training-validation-testing sets. The MIMIC-CXR [17] includes 377,110 chest x-ray images associated with 227,835 reports. The dataset is officially split into 368,960 images (222,758 reports) for training, 2,991 images (1,808 reports) for validation and 5,159 images (3,269 reports) for testing. It is worth noting that we focus on the unsupervised medical report generation, where the image-report pairs are not available, thus the image-report training pairs of both the IU X-ray and MIMIC-CXR datasets are discarded and are not used in our unsupervised training stage. Only the training reports of MIMIC-CXR and IU X-ray, i.e., 222,758 + 2,770 = 225,528 reports, are used as the independent report corpus to train our unsupervised model. Only under the (semi-)supervised training setting, we will adopt the image-report training pairs to train our approach.

**Metrics** To fairly compare with existing models [25, 6], we adopt the evaluation toolkit [5] to calculate the widely-used natural language generation metrics, i.e., BLEU [34], METEOR [3] and ROUGE-L [26], which measure the match between the generated reports and ground truth reports, but are not specialized for the abnormalities in the reports. Therefore, to measure the accuracy of descriptions for clinical abnormalities, we further report clinical efficacy metrics following the work of Chen et al. [6]. The clinical efficacy metrics are calculated by comparing the generated reports with ground truth reports in 14 different categories related to thoracic diseases and support devices, producing the Precision, Recall and F1 scores.

**Settings** The size $d$ is set to 256. For the attention mechanism in Eq. (4) and Eq. (5), we adopt the multi-head attention [38], the number of heads $n$ in multi-head attention is set to 8. The intermediate dimension in $\mathcal{F}$, i.e., Eq. (4), is set to 1024. Based on the average performance on the validation set, the $N_B$ in knowledge bank, i.e., Eq. (6), is set to 10,000. The $N_{KG}$ in the knowledge graph is set to 200. In our knowledge-driven encoder, the image embedding module adopts the ResNet-50 [12] pre-trained on ImageNet [10] and fine-tuned on CheXpert dataset [14] to extract the image embeddings in the shape of $7 \times 7 \times 2048$, which will be projected to $d = 256$, acquiring $I' \in \mathbb{R}^{49 \times 256}$, i.e., $N_I = 49$; The report embedding module is implemented by the Transformer [38] equipping with the self-attention mechanism provided in Lin et al. [27]; $N_R = N_I = 49$. In both unsupervised and (semi-)supervised settings (see Section 3.3), the batch size is set to 16 and Adam optimizer [18] with a learning rate of 1e-4 is used for parameter optimization. Before testing on the IU X-ray/MIMIC-CXR, we further only employ the training data from IU X-ray/MIMIC-CXR to train the model. All re-implementations and our experiments were run on 8 V100 GPUs.

### 4.2 Automatic Evaluation

We evaluate the performance of our approach under unsupervised, semi-supervised and supervised settings. The results are shown in Table 1, Table 2 and Figure 2. We select several supervised methods, including a recently state-of-the-art model R2Gen [6], for comparison. These models follow the encoder-decoder architecture, trained on the full pairs of images and reports.

**Unsupervised Setting** As shown in Table 1 and Table 2, our unsupervised model KGAE achieves competitive results with some supervised models in both IU X-ray and MIMIC-CXR datasets, and

Table 1: Performance in terms of natural language generation metrics on the IU X-ray and MIMIC-CXR. B-n, M and R-L are short for BLEU-n, METEOR and ROUGE-L, respectively. Higher is better in all columns. It is worth noting that the KGAE is trained in an unsupervised manner, KGAE-Semi is trained 60% paired data, KGAE-Supervised and existing methods are trained with full paired data.

| Methods | Year | Ratio of Pairs | IU X-ray [9] | | | | | | MIMIC-CXR [17] | | | | | |
|---|---|---|---|---|---|---|---|---|---|---|---|---|---|---|
| | | | B-1 | B-2 | B-3 | B-4 | M | R-L | B-1 | B-2 | B-3 | B-4 | M | R-L |
| NIC [39] | 2015 | 100% | 0.216 | 0.124 | 0.087 | 0.066 | - | 0.306 | 0.299 | 0.184 | 0.121 | 0.084 | 0.124 | 0.263 |
| AdaAtt [31] | 2017 | 100% | 0.220 | 0.127 | 0.089 | 0.068 | - | 0.308 | 0.299 | 0.185 | 0.124 | 0.088 | 0.118 | 0.266 |
| Att2in [35] | 2017 | 100% | 0.224 | 0.129 | 0.089 | 0.068 | - | 0.308 | 0.325 | 0.203 | 0.136 | 0.096 | 0.134 | 0.276 |
| Transformer [6] | 2020 | 100% | 0.396 | 0.254 | 0.179 | 0.135 | 0.164 | 0.342 | 0.314 | 0.192 | 0.127 | 0.090 | 0.125 | 0.265 |
| $\mathcal{M}^2$ Trans. [7] | 2020 | 100% | 0.437 | 0.290 | 0.205 | 0.152 | 0.176 | 0.353 | 0.238 | 0.151 | 0.102 | 0.067 | 0.110 | 0.249 |
| R2Gen [6] | 2020 | 100% | 0.470 | 0.304 | 0.219 | 0.165 | 0.187 | 0.371 | 0.353 | 0.218 | 0.145 | 0.103 | 0.142 | 0.277 |
| KGAE | | 0% | 0.417 | 0.263 | 0.181 | 0.126 | 0.149 | 0.318 | 0.221 | 0.144 | 0.096 | 0.062 | 0.097 | 0.208 |
| KGAE-Semi | Ours | 60% | 0.497 | 0.320 | 0.232 | 0.171 | 0.189 | 0.379 | 0.352 | 0.219 | 0.149 | 0.108 | 0.147 | 0.290 |
| KGAE-Supervised | | 100% | **0.512** | **0.327** | **0.240** | **0.179** | **0.195** | **0.383** | **0.369** | **0.231** | **0.156** | **0.118** | **0.153** | **0.295** |

Table 2: Performance of automatic evaluation in terms of clinical efficacy metrics, which measure the accuracy of descriptions for clinical abnormalities, on the test set of MIMIC-CXR dataset. Higher is better in all columns.

| Methods | Year | Ratio of Pairs | MIMIC-CXR [17] | | |
|---|---|---|---|---|---|
| | | | Precision | Recall | F1 |
| NIC [39] | 2015 | 100% | 0.249 | 0.203 | 0.204 |
| AdaAtt [31] | 2017 | 100% | 0.268 | 0.186 | 0.181 |
| Att2in [35] | 2017 | 100% | 0.322 | 0.239 | 0.249 |
| Up-Down [1] | 2018 | 100% | 0.320 | 0.231 | 0.238 |
| $\mathcal{M}^2$ Trans. [7] | 2020 | 100% | 0.197 | 0.145 | 0.133 |
| Transformer [6] | 2020 | 100% | 0.331 | 0.224 | 0.228 |
| R2Gen [6] | 2020 | 100% | 0.333 | 0.273 | 0.276 |
| KGAE | | 0% | 0.214 | 0.158 | 0.156 |
| KGAE-Semi | Ours | 60% | 0.360 | 0.302 | 0.307 |
| KGAE-Supervised | | 100% | **0.389** | **0.362** | **0.355** |

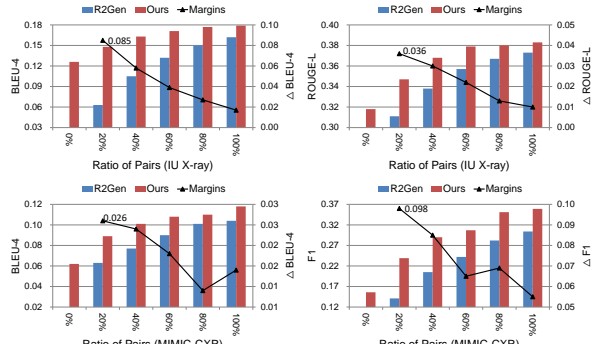

Figure 2: Performance of re-implemented state-of-the-art model R2Gen [6] and our approach on the test sets with respect to various amount of paired data used for training. The margins in different ratios are shown with the polyline and the right y-axis. As we can see, our approach consistently outperforms the R2Gen.

even outperforms several supervised models. Specifically, on the IU X-ray dataset, Table 1 shows that KGAE surpasses the NIC [44], AdaAtt [31] and Att2in [35] in terms of all metrics, and the Transformer [6] in terms of BLEU-1,2,3. On the MIMIC-CXR dataset, Table 2 shows that KGAE outperforms the $\mathcal{M}^2$ Trans. [7] in terms of Precision, Recall and F1. The competitive results prove the effectiveness of our approach in addressing the unsupervised medical report generation, and thus can generate desirable medical reports without the training on the pairs of image and report.

**Semi-Supervised Setting** To further prove the effectiveness of our approach, we fine-tune the unsupervised KGAE using partial downstream paired image-report datasets (see Section 3.3), resulting in the KGAE-Semi. To this end, in Figure 2, we evaluate the performance of our approach on both IU X-ray and MIMIC-CXR datasets with respect to the increasing amount of paired data. For a fair comparison, we also re-train the state-of-the-art model R2Gen [6] using the same amount of pairs. As we can see, our model outperforms the R2Gen under all ratios of paired dataset used for training. It is worth noting that the fewer the image-report pairs, the larger the margins, e.g., under the very limited pairs setting (20% of paired datasets), our approach significantly surpasses the R2Gen by 8.5% absolute BLEU-4 score on IU X-ray and 9.8% absolute F1 score on MIMIC-CXR. Intuitively, since our approach can relax the reliance on the paired datasets, we can make use of available unpaired image and report data as a solid bias for medical report generation task. Table 1 and Table 2 further prove the effectiveness of our approach, which achieves results competitive with current state-of-the-art models by using only 60% of paired dataset.

**Supervised Setting** We fine-tune the unsupervised KGAE using full image-report pairs, acquiring the KGAE-Supervised model (see Section 3.3). Table 1 and Table 2 show that KGAE-Supervised sets the new state-of-the-art results on the two datasets in all metrics. Moreover, in terms of the clinical

Table 3: We invite three professional clinicians to conduct human evaluation for comparing our approach with $\mathcal{M}^2$ Trans. [7] and R2Gen [6] under various amount of pairs for training in terms of the *comprehensiveness* of the generated true abnormalities and the *faithfulness* of the generated normalities and abnormalities to the ground truth reports. All values are reported in percentage (%).

| Metrics | KGAE (0%) vs. $\mathcal{M}^2$ Trans. (100%) | | | KGAE (0%) vs. R2Gen (100%) | | | KGAE-Semi (20%) vs. R2Gen (20%) | | | KGAE-Supervised (100%) vs. R2Gen (100%) | | |
|---|---|---|---|---|---|---|---|---|---|---|---|---|
| | Loss | Tie | Win | Loss | Tie | Win | Loss | Tie | Win | Loss | Tie | Win |
| Faithfulness | 34 | 18 | **48** | **62** | 15 | 23 | 21 | 10 | **69** | 25 | 22 | **53** |
| Comprehensiveness | 27 | 20 | **53** | **57** | 17 | 26 | 24 | 11 | **65** | 32 | 21 | **47** |

efficacy metrics, our approach achieves 0.389 precision score, 0.362 recall score and 0.355 F1 score, outperforming the state-of-the-art model R2Gen [6]. The superior clinical efficacy scores demonstrate the capability of our approach to produce higher quality descriptions for clinical abnormalities than existing models.

**Overall** Combining the results of unsupervised, semi-supervised and supervised settings, the proposed KGAE can relax the dependency on the paired datasets, and thus makes the medical report generation model use the available separate image and report data to boost the performance. The advantages under the scenarios with limited labeled pairs (i.e., semi-supervised setting) show that KGAE might be applied to other medical images (MRI, Dermoscopy, Retinal, etc.), where the coupled images and reports pairs could be much less or even unavailable.

### 4.3 Human Evaluation

We conduct human evaluations to verify the effectiveness of KGAE in clinical practice. Specifically, to assist radiologists in clinical decision-making and reduce their workload, it is important to generate accurate reports (*faithfulness*), i.e., the model does not generate normalities and abnormalities that does not exist according to doctors, with comprehensive abnormalities (*comprehensiveness*), the model does not leave out the abnormalities. Therefore, we randomly select 100 samples from the MIMIC-CXR and invite three professional clinicians to compare our approach and baselines independently. The clinicians are unaware of which model generates these reports. The results are shown in Table 3. As we can see, under the unsupervised setting, our approach achieves competitive results with the supervised model, outperforming the $\mathcal{M}^2$ Trans with winning pick-up percentages. Under the (semi-)supervised setting, our method is better than state-of-the-art model R2Gen [6] in all metrics, especially for the semi-supervised setting (20% of paired dataset), our method substantially surpasses the R2Gen, which is in accordance with the automatic evaluation, by $69 - 21 = 48$ and $65 - 24 = 41$ points in terms of the *faithfulness* and *comprehensiveness* metrics, respectively.

## 5 Analysis

In this section, we conduct several analysis to better understand our proposed approach.

### 5.1 Knowledge Graph Sensitivity

In this section, to evaluate the knowledge graph sensitivity, we evaluate the performances using different knowledge graphs defined on IU X-Ray only, MIMIC-CXR only, both MIMIC-CXR and IU X-Ray. Table 4 shows the results of KGAE (0%) and KGAE-Supervised (100%) on the IU X-ray dataset. As we can see, our KGAE using different knowledge graphs can consistently outperform several existing supervised models, i.e., NIC, AdaAtt, Att2in, across all metrics (Table 1). Similarly, our KGAE-Supervised with different knowledge graphs can also consistently outperform existing state-of-the-art model, i.e., R2Gen (Table 1). The results prove the robustness of our proposed model to the pre-defined knowledge graph. Therefore, this work could provide a good basis or starting point for the research of unsupervised medical report generation in other clinical domains such as MRI and Dermoscopy.

### 5.2 Ablation Study

In Table 5, we conduct quantitative analysis to better understand our approach under both the unsupervised and supervised training settings. For different ablation settings, the KGAE-Supervised is acquired by further fine-tuning the KGAE using image-report pairs.

Table 4: Analysis of the knowledge graph sensitivity. Performance of our approach using different knowledge graphs defined on IU X-Ray only, MIMIC-CXR only, both MIMIC-CXR and IU X-Ray.

| Methods | Ratio of Pairs | Knowledge Graphs | IU X-ray [9] | | | | | |
|---|---|---|---|---|---|---|---|---|
| | | | B-1 | B-2 | B-3 | B-4 | M | R-L |
| KGAE | 0% | IU X-Ray | **0.425** | **0.271** | **0.185** | 0.114 | 0.123 | 0.310 |
| | | MIMIC-CXR | 0.417 | 0.263 | 0.181 | **0.126** | **0.149** | 0.318 |
| | | MIMIC-CXR + IU X-Ray | 0.419 | 0.260 | 0.180 | 0.124 | 0.143 | **0.321** |
| KGAE-Supervised | 100% | IU X-Ray | **0.519** | **0.331** | 0.235 | 0.174 | 0.191 | 0.376 |
| | | MIMIC-CXR | 0.512 | 0.327 | **0.240** | 0.179 | 0.195 | 0.383 |
| | | MIMIC-CXR + IU X-Ray | 0.505 | 0.323 | 0.239 | **0.181** | **0.198** | **0.385** |

Table 5: We conduct the quantitative analysis under both the unsupervised setting (KGAE) and supervised setting (KGAE-Supervised). We analysis the effect of the shared $\mathcal{F}$ in the encoder (Eq. (4)), and the effect of the number of the knowledge in the decoder's knowledge bank $B$ (Eq. (6)).

| Methods | Settings | $\mathcal{F}$ | $B$ | IU X-ray [9] | | | | | | MIMIC-CXR [17] | | | | | |
|---|---|---|---|---|---|---|---|---|---|---|---|---|---|---|---|
| | | | | B-1 | B-2 | B-3 | B-4 | M | R-L | B-1 | B-2 | B-3 | B-4 | M | R-L |
| KGAE (0%) | (a) | - | - | 0.291 | 0.170 | 0.127 | 0.086 | 0.120 | 0.301 | 0.166 | 0.095 | 0.064 | 0.032 | 0.070 | 0.174 |
| | (b) | √ | - | 0.352 | 0.227 | 0.154 | 0.109 | 0.133 | 0.313 | 0.192 | 0.118 | 0.079 | 0.045 | 0.085 | 0.187 |
| | (c) | √ | 5,000 | 0.403 | 0.252 | 0.170 | 0.117 | 0.144 | 0.316 | 0.211 | 0.137 | 0.093 | 0.058 | 0.093 | 0.202 |
| | (d) | √ | 15,000 | 0.412 | 0.260 | 0.178 | 0.121 | 0.148 | **0.320** | 0.217 | 0.140 | 0.092 | 0.059 | **0.101** | 0.205 |
| | Full Model | √ | 10,000 | **0.417** | **0.263** | **0.181** | **0.126** | **0.149** | 0.318 | **0.221** | **0.144** | **0.096** | **0.062** | 0.097 | **0.208** |
| KGAE-Supervised (100%) | (e) | - | - | 0.482 | 0.310 | 0.219 | 0.166 | 0.184 | 0.373 | 0.323 | 0.204 | 0.128 | 0.104 | 0.115 | 0.266 |
| | (f) | √ | - | 0.470 | 0.304 | 0.217 | 0.162 | 0.181 | 0.368 | 0.320 | 0.206 | 0.129 | 0.101 | 0.118 | 0.267 |
| | (g) | √ | 5,000 | 0.505 | 0.323 | 0.232 | 0.178 | 0.190 | 0.379 | 0.363 | 0.228 | 0.152 | 0.115 | 0.147 | 0.290 |
| | (h) | √ | 15,000 | 0.498 | 0.315 | 0.221 | 0.170 | 0.186 | 0.374 | 0.368 | **0.235** | **0.158** | 0.114 | 0.150 | 0.293 |
| | Full Model | √ | 10,000 | **0.512** | **0.327** | **0.240** | **0.179** | **0.195** | **0.383** | **0.369** | 0.231 | 0.156 | **0.118** | **0.153** | **0.295** |

As we can see, under the unsupervised setting, both the introduced shared $\mathcal{F}$ and knowledge bank $B$ can significantly boost the performance, which proves our arguments and verifies the effectiveness of our approach in performing the unsupervised medical report generation.

Under the supervised setting, as shown in settings (e,f), applying shared $\mathcal{F}$ generates unchanged and impaired performance on the MIMIC-CXR and IU X-ray datasets, respectively. We speculate the reason is that the supervised model no longer requires the $\mathcal{F}$ to bridge the vision and the language domains. Therefore, the performance is unchanged on the large dataset MIMIC-CXR. However, the increased parameters introduced by $\mathcal{F}$ might bring overfitting or increase the difficulty in optimization on the small dataset IU X-ray, which somewhat hinders the performance. For the knowledge bank $B$, we can find that the results of setting (h) outperforms (g) on the MIMIC-CXR dataset, but underperforms (g) on the IU X-ray dataset. We speculate the reason is that the large MIMIC-CXR dataset contains more knowledge than the small IU X-ray dataset, so a larger knowledge bank may learn more knowledge of MIMIC-CXR to boost the performance while introducing more noisy knowledge into the IU X-ray dataset to degrade the performance.

## 5.3 Qualitative Analysis

In Figure 3, we conduct the qualitative analysis to better understand our approach. As we can see, the visualization verifies the effectiveness of our knowledge-driven encoder in extracting the knowledge representations of both image and report. For the generated reports, our unsupervised KGAE generates a desirable report, which correctly describes "*innumerable nodules are present*" and "*heart size is normal*". When removing the bank $B$, the model tends to generate plausible general reports with no prominent abnormal narratives and some repeated reports, which shows that the knowledge memory mechanism can indeed distill and preserve the desired medical knowledge to boost the generation of reports. Under the semi-supervised setting, the R2Gen can not well handle the medical report generation task and generates some repeated sentences of normalities (Underlined text) and fails to depict some rare but important abnormalities, i.e., "*nodules*" and "*scoliosis*", while our approach can generate fluent report supported by accurate abnormalities. Under the supervised setting, our approach can generate an accurate report showing significant alignment with the ground truth report. It further prove our arguments and the effectiveness of our proposed approach.

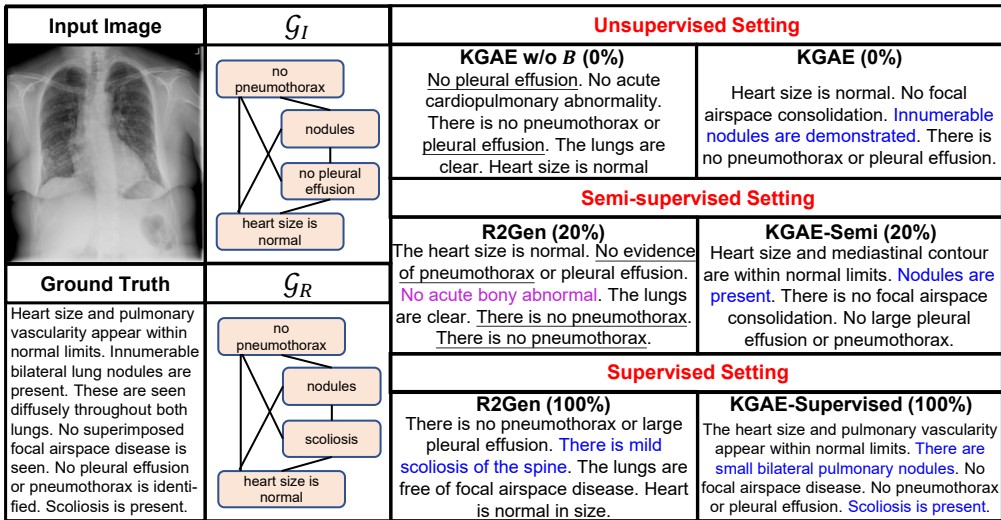

Figure 3: Reports generated by our approach and a state-of-the-art model R2Gen [6]. Under the unsupervised setting, we show four nodes with top-4 attention weights in Eq. (4) to visualize $\mathcal{G}_I$ and $\mathcal{G}_R$. The Purple and Blue colored text denote the generated wrong sentences (*Faithfulness*) and true abnormalities (*Comprehensiveness*), respectively; Underlined text denotes the repeated sentences.

## 6   Conclusions and Discussions

In this paper, we propose the Knowledge Graph Auto-Encoder (KGAE). Without any image-report pairs, KGAE can extract the knowledge representations of both image and report from the knowledge graph to bridge the visual and textual domains, and generate desirable reports by being trained in the auto-encoding pipeline. The experiments verify the effectiveness of our approach, which even exceeds several supervised models. Moreover, by further fine-tuning KGAE using paired datasets, we achieve the state-of-the-art results on two public datasets with the best human preference.

In the future, 1) since we can relax the dependency on paired data, it can be interesting to apply the KGAE to other types of medical images of other body parts, where the image-report pairs could be much less or even unavailable, to assist radiologists in clinical decision-making and reduce their workload; 2) We can replace the explicit pre-defined knowledge graph with an implicit large matrix (e.g., knowledge bank in our decoder) to improve the generalization ability of our approach.

**Societal Impacts:** In this paper, we target the problem of medical report generation. Although the proposed model outperforms state-of-the-art approaches, it aims to assist the radiologists instead of replacing them. For experienced radiologists, given a large amount of medical images, our model can automatically generate medical reports, the radiologists only need to make revisions rather than write a new report from scratch. However, it is possible that some radiologists direct copy the generated report as the final report. Also for less experienced radiologists, they may not be able to correct the errors in machine-generated reports. In order to apply the proposed model in clinical practice, it is required to add process control to avoid unintended use.

**Limitations:** Although the proposed KGAE can work in an unsupervised manner, we still need the independent sets of medical images and medical reports which may still be difficult to collect for some types of medical images. In addition, our approach introduces the knowledge graph to bridge visual and textual domain, in the paper, we collect the frequent clinical findings as nodes and build the knowledge graph from the set of reports automatically. When applying to new domains, we need to collect a new set of clinical findings.

## Acknowledgments

We would like to sincerely thank the clinicians (Xiaoxia Xie, Jing Zhang, etc.) of the Harbin Chest Hospital in China for providing the human evaluation. We also sincerely thank all the anonymous reviewers and chairs for their constructive comments and suggestions that substantially improved this paper. Xu Sun and Sheng Wang are the corresponding authors of this paper.

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
