# OpenReview forum: "Auto-Encoding Knowledge Graph for Unsupervised Medical Report Generation"
_NeurIPS.cc/2021/Conference — NeurIPS 2021 Poster_

### Official Review · Reviewer_S5sP · 2021-07-03

**Rating:** 5
**Confidence:** 3

**Summary:**

For automatic medical report generation, this work 1) builds a knowledge graph from the report corpus of MIMIC-CXR, 2) designs a knowledge-driven encoder which uses a ResNet-50 and a BiLSTM trained in a multi-label classification task, 3) designs a knowledge-driven decoder based on Transformer and memory network. The proposed model doesn’t require any image-report training pairs, in other words, it uses independent sets of images and reports in training, but it outperforms several supervised models on MIMIC-CXR and IU X-ray datasets. This method can be further extended to semi-supervised and supervised settings, and outperforms current SOTA.

**Limitations And Societal Impact:**

It is not clear how both graphs share the common latent space without training text-image together. Further analysis is needed to prove that the knowledge encoder can indeed encode images and reports to a common latent space.

The unsupervised method, which is the main contribution of this work, does not outperform SOTA.

The knowledge graph is built from the training set of MIMIC-CXR, hence not applicable to other clinical domains.

Minor:

The edge weights in the knowledge graph are based on normalized co-occurrence. Sometimes co-occurrence may not represent the correct relations, for example in the case of negations. Maybe a more sophisticated definition of edge weights can further boost the performance.

Page 1, line 36. ‘labeled’ seems misleading.


**Main Review:**

This paper embeds the image using a trained ResNet-50, attends the embedding to the knowledge graph embeddings, and finally obtains an image knowledge representation. The same process applies to reports, through a trained BiLSTM. In this way, the paper obtains the attended knowledge related to the image and report, and they (image, report knowledge) share the common latent space, which allows us to train the model using independent sets of images and reports.

The paper is well-written with no obvious errors. The motivation and method descriptions are clear.

The knowledge graph nodes are clinical abnormalities and normalities, which can be in the form of phrases, instead of only keywords.

Proposed knowledge-driven attention incorporates a knowledge bank to implement a knowledge memory mechanism, which is similar to a memory network. In this way, the input knowledge representation can distill the fine-grained knowledge preserved in the knowledge bank for better accuracy. The bank can be learned in an end-to-end manner.

Besides the widely-used NLG metrics, the paper also uses clinical efficacy metrics to evaluate the accuracy of descriptions of clinical abnormalities.




**Time Spent Reviewing:**

1 hour

---

> ### Author Response · Authors · 2021-08-10
> **Response to Reviewer@S5sP**
>
> Thanks for your helpful comments! If you have further concerns, please feel free to contact us.
>
> > **Q1**: It is not clear how both graphs share the common latent space without training text-image together.
>
> **A1**: Note that in Equation (4) of our paper, we further introduce a shared $\mathcal{F}$ to encode the obtained image and report knowledge representations, and try to project both representations to the same latent space. To prove our arguments, in Figure 3, we adopt the t-SNE to visualize the image and report knowledge representations. We can find that there is a large gap between the raw image and report embeddings, while the encoded image and report knowledge representations better share the common latent space. The analysis proves that our proposed knowledge encoder can indeed encode images and reports to a common latent space (L309-312).
>
> > **Q2**: The unsupervised method does not outperform SOTA.
>
> **Q2**: Although the proposed unsupervised method, which does not require any image-report training pairs, does not outperform SOTA supervised models, it can generate desirable medical reports (please see Figure 3 of our paper) and can outperform some supervised models, e.g., NIC, AdaAtt and Att2in (please see Table 1 of our paper). Therefore, the relaxation of the reliance on paired training data enlarges the application scope of our proposed unsupervised model. In addition, under the supervised setting, our proposed model can outperform SOTA approaches (please see Table 1 of our paper).
>
> > **Q3**: The knowledge graph (KG) is built from the training set of MIMIC-CXR, hence not applicable to other clinical domains.
>
> **A3**: To address your concern, we attempt to evaluate the performances using different KGs defined on IU X-Ray only, MIMIC-CXR only, both MIMIC-CXR and IU X-Ray. Two tables below show the results of our unsupervised model KGAE (0%) and supervised model KGAE-Supervised (100%) on the IU X-ray dataset:
>
> | Methods | Ratio of Pairs | KGs | BLEU-1  | BLEU-2 | BLEU-3 | BLEU-4 | METEOR | ROUGE-L |
> | ---------- | :-----------:  | ----------- | :-----------: | :-----------: | :-----------: | :-----------: | :-----------: | :-----------: |
> | NIC [38] | 100% | - |0.216 | 0.124 | 0.087 | 0.066 | - | 0.306 |
> | AdaAtt [29] | 100% | - | 0.220 | 0.127 | 0.089 | 0.068 | - | 0.308 |
> | Att2in [33] | 100% | - | 0.224 | 0.129 | 0.089 | 0.068 | - | 0.308 |
> | |
> | KGAE (0%) [Ours] | 0% | IU X-Ray | **0.425** | **0.271** | **0.185** | 0.114 | 0.123 | 0.310 |
> | KGAE (0%) [Ours] | 0% | MIMIC-CXR  | 0.417 | 0.263 | 0.181 | **0.126** | **0.149** | 0.318 |
> | KGAE (0%) [Ours] | 0% | MIMIC-CXR + IU X-Ray | 0.419 | 0.260 | 0.180 | 0.124 | 0.143 | **0.321** |
> | |
>
> | Methods | Ratio of Pairs |KGs | BLEU-1  | BLEU-2 | BLEU-3 | BLEU-4 | METEOR | ROUGE-L |
> | ---------- | :-----------:  | -----------  | :-----------: | :-----------: | :-----------: | :-----------: | :-----------: | :-----------: |
> | R2Gen [6] | 100% | - | 0.470 | 0.304 | 0.219 | 0.165 | 0.187 | 0.371 |
> | PPKED [28] | 100% | - | 0.483 | 0.315 | 0.224 | 0.168 | 0.190 | 0.376 |
> | |
> | KGAE-Supervised (100%) [Ours] | 100% | IU X-Ray | **0.519** | **0.331** | 0.235 | 0.174 | 0.191 | 0.376 |
> | KGAE-Supervised (100%) [Ours] | 100% | MIMIC-CXR | 0.512 | 0.327 | **0.240** | 0.179 | 0.195 | 0.383 |
> | KGAE-Supervised (100%) [Ours] | 100% | MIMIC-CXR + IU X-Ray | 0.505 | 0.323 | 0.239 | **0.181** | **0.198** | **0.385** |
> | |
>
> As we can see, our KGAE (0%) using different knowledge graphs can consistently outperform several existing supervised models, i.e., NIC, AdaAtt, Att2in, across all metrics. Similarly, our KGAE-Supervised (100%) with different knowledge graphs can also consistently outperform existing state-of-the-art models, i.e., R2Gen and PPKED. The results prove the robustness of our proposed model to the knowledge graph.
>
> Besides, we build the knowledge graph with the following two steps: 1) first acquire a dictionary of key terms as nodes; 2) initialize the weights between terms according to the occurrence frequencies. These two steps don't need manual efforts.
>
> Therefore, this work could provide a good basis or starting point for the research of unsupervised medical report generation. The good robustness of our proposed model to the knowledge graph, which doesn't need manual efforts to build, enables our method to have the potential to be applied to other clinical domains such as MRI and Dermoscopy, where the image-report pairs could be much less or even unavailable.
>
> > **Q4**: Maybe a more sophisticated definition of edge weights can further boost the performance.
>
> **A4**: Thank you for the great suggestion. We agree with your point that a more sophisticated definition of edge weights may further boost the performance. It is worth noticing that we have adopted a graph convolution network (GCN) to embed the knowledge graph, where the edge weights will be adaptively updated (i.e., learned) by GCN during the training process. Thus, after GCN iterations, different definitions of edge weights may not affect the performance much.
>
> To prove our argument, we further evaluate the performance of using the knowledge graph, where the initial edge weights are all set to 1. The results on the IU X-ray dataset are reported in the following Table:
>
> | Edge Weights | BLEU-1  | BLEU-2 | BLEU-3 | BLEU-4 | METEOR | ROUGE-L |
> | ---------- | :-----------:  | :-----------: | :-----------: | :-----------: | :-----------: | :-----------: |
> | Set to 1 |  0.385 | 0.238 | 0.154 | 0.107 | 0.134 | **0.325** |
> | Normalized Co-occurrence (Our KGAE (0%))  | **0.417** | **0.263** | **0.181** | **0.126** | **0.149** | 0.318|
> ||
>
> The results show that the performance using the knowledge graph, where the initial edge weights are all set to 1, is competitive with the KGAE (0%).
>
> > **Q5**: Page 1, line 36. 'labeled' seems misleading.
>
> **A5**: Thank you! The 'labeled' means that the medical report generation datasets can only be manually labeled by professional radiologists, we will clarify the statement here.

---

### Official Review · Reviewer_bvmd · 2021-07-15

**Rating:** 6
**Confidence:** 5

**Summary:**

This manuscript proposes an unsupervised method to generate medical report (a paragraph of textual description of medical image) for a given unlabelled medical image. The scenario is interesting but I have several major concerns for the model, as following comments.

**Limitations And Societal Impact:**

The societal impact provided by the authors is ok for me. I don't have particular suggestions.

**Main Review:**

1. Why use knowledge graph, is it really necessary? I saw the motivation in the beginning of Section 3.2, but it's not convincing. Please justify it better and provide a simple comparison experiment: replace the graph by CNN, see how it works. In detail, only keep the node feature matrix V (Line 110; ignore the edges E) and use CNN to process it to get embedding v_i of each ab/normalities.

BTW, the notation of the knowledge graph is inappropriate. In line 110, the notation V denotes node features (each node has d-dimension feature) but not 'a set of nodes'. Please check some graph related literature and improve the notation.

2. How many KE are used in this model? one or two? Are KE_I and KE_R share all parameters or partial parameters?

3. The learned embedding I' and R' are not under the same distribution although they are in the same latent space. I suggest to add a mapping function (e.g., a Fully-connected layer) on I' and R', respectively, to map them into the same distribution.

4. A major issue: Line 145, if the input is an image, there is no word, so what's w_t? Or will this model still appliable?

5. The concept of knowledge bank (knowledge memory mechanism) is interesting to me. How to get the bank B, is it pre-defined or learnable?

6. Line 173-175, how to get the label of these images or reports in multi-label classification? Suppose the dataset itself contains these labels, it still makes this whole framework a semi-supervised model? At least not a strict unsupervised framework. I understand this issue is not easy to address in this work, please discuss this limitation in Discussion.


Minor comments:

- In fig 1, since green line denotes training and red line denotes testing, it seems that the model only accepts report in training and only accept images in testing. But the model will take both images and reports in training, right? Please revise the figure accordingly.

- Please add motivation of the use of knowledge graph in Introduction. I saw the motivation in Section 3.2, but it's too far behind.


---------------Update after rebuttal------------------
Thank the authors for the detailed response. The answers addressed most of my concerns and I boosted my score from 5 to 6.

**Time Spent Reviewing:**

2

---

> ### Author Response · Authors · 2021-08-10
> **Response to Reviewer@bvmd**
>
> Thanks for your helpful comments! If you have further concerns, please feel free to contact us.
>
> > **Q1**: The motivation of using knowledge graph (KG).
>
> **A1**: In our work, we propose the unsupervised medical report generation model, thus, a key challenge is to bridge the gap between vision and language domains without training on the pairs of images and reports. To this end, we adopt the knowledge graph to create a latent space and propose a knowledge-driven encoder, which includes a common mapping function, to project images and reports to the same latent space (Line 50-55). As a result, our encoder can extract the image and report knowledge representations, i.e., the knowledge related to the image and report, they (image, report knowledge) share the common latent space, which allows our model to bridge the gap between vision and language domains without the training on the pairs of image and report.
>
> The proposed approach already includes a CNN model, in Line 126-128 and Line 227-230, the image embedding I' is acquired by feeding the image through the CNN model. The knowledge-driven encoder is equipped following the CNN model to transform I' to $\mathcal{G}_I$. The approach you mentioned that keeps the node feature matrix V (Line 110; ignore the edges E) is similar to PPKED [28], as shown in Table 1 of our paper, under the supervised setting, our KGAE-Supervised outperforms PPKED.
>
> For the notation of the knowledge graph, we will follow your advice to revise.
>
> > **Q2**: How many KE are used in this model? one or two? Are KE_I and KE_R share all parameters or partial parameters?
>
> **A2**: As shown in Eq. (4), we adopt two KEs, i.e., KE_I and KE_R, and each KE includes an attention model as well as a commonly shared $\mathcal{F}$. The only shared weights of KE_I and KE_R are the parameters of $\mathcal{F}$, and the parameters of attention models are independent.
>
> > **Q3**: Add a mapping function (e.g., a Fully-connected layer) on I' and R', respectively, to map them into the same distribution.
>
> **A3**: We agree with you that a mapping function is necessary to map I' and R' into the same distribution, thus, in our model, as shown in Eq. (4) and Line 138-139, we have incorporated such mapping function, i.e., the shared $\mathcal{F}$, which is implemented as two fully-connected (FC) layers with a ReLU in between (i.e., FC-ReLU-FC).
>
> We compare our unsupervised model KGAE (0%) with KGAE (0%) w/o $\mathcal{F}$ under the unsupervised setting and find $\mathcal{F}$ is critical to the performance:
>
> | Methods | BLEU-1  | BLEU-2 | BLEU-3 | BLEU-4 | METEOR | ROUGE-L |
> | ---------- | :-----------:  | :-----------: | :-----------: | :-----------: | :-----------: | :-----------: |
> | KGAE (0%) | **0.417** | **0.263** | **0.181** | **0.126** | **0.149** | **0.318** |
> | w/o $\mathcal{F}$ | 0.265 | 0.151 | 0.105 | 0.074 | 0.097 | 0.268 |
> | |
>
> > **Q4**: Line 145, if the input is an image, there is no word, so what's w_t? Or will this model still be applicable?
>
> **A4**: In Line 145, we describe the details of the language decoder, which is widely used for text generation in the vision-to-text task. As a language decoder, the input and output are all text, here the w_t is always a word. If the input is an image, as shown in Eq. (5), the w_t will be transformed to h_t and the h_t will work together with the visual features from the input image to generate the next word.
>
> > **Q5**: How to get the bank B, is it pre-defined or learnable?
>
> **A5**: Thank you for your interest in our knowledge bank B! The knowledge bank B is composed of N_B  learnable vectors, which are randomly initialized and are learned from scratch during training.
>
> > **Q6**: Line 173-175, how to get the label of these images or reports in multi-label classification?
>
> **A6**: We adopt the dataset CheXpert for image classification and MIMIC-CXR for report classification, respectively. Note that there are no paired image-report samples between CheXpert and MIMIC-CXR. For the labels of images and reports, they adopt the same 14 labels, which are the widely-used 14 common radiographic observations in the research of chest X-rays. The images of the CheXpert dataset are already labeled, like the widely-used ImageNet dataset. For the labels of reports from MIMIC-CXR, we adopt the CheXpert NLP Labeler (https://github.com/MIT-LCP/mimic-cxr/tree/master/txt/chexpert) to extract the 14 common radiographic observations from textual reports.
>
> > **Q7**: Figure 1.
>
> **A7**: Thank you! In Figure 1, the green and red lines respectively denote the training and testing stages in the report generation process, i.e., the decoder. In training the decoder, it only accepts reports. In predicting the reports with the decoder, it only accepts images. However, in training the encoder, it accepts both images and reports, but these images and reports can be independent. We will follow your great advice to revise the figure accordingly and provide a detailed explanation of Figure 1 in our revision.

---

### Official Review · Reviewer_AGjp · 2021-07-17

**Rating:** 6
**Confidence:** 5

**Summary:**

This paper proposes a knowledge graph based method for medical report generation which can be trained by the unsupervised、semi-supervised and supervised manner. The overall framework consists of a pre-constructed knowledge graph、a knowledge-driven encoder and a knowledge-driven decoder. The authors first construct a medical knowledge graph and embed it by a graph embedding module, after that, they use the knowledge-driven encoder to obtain the knowledge representation of the image (report) by calculating the self-attention of the knowledge graph embeddings and the image (report) embeddings,  and then feed the knowledge representation into the knowledge-driven decoder to generate the report. As a result, the proposed framework outperforms several strong baselines on two datasets, i.e., IU-Xray and MIMIC-CXR.

**Limitations And Societal Impact:**

The current paper is limited in the following aspects.

1、	This paper lacks the clarification about the new contribution of the proposed knowledge graph with respect to those knowledge-graph based medical report generation methods (except the proposed method is an unsupervised model), such as [47, 22] mentioned in the related works. There also lacks of experimental comparison with these methods.

2、	This paper lacks the details of how to construct the knowledge graph. For example, In line 111-114, the authors said they consider the terms of the clinical abnormalities and normalities as nodes: are those nodes are selected manually?

3、	The authors should explain more why the unsupervised training method can work. For example, In line 173-177, the authors train a common multi-label classification network with knowledge representation of images and reports, how can this bridge the vision and the language domains?

4、	In the experimental validation, the commonly used criteria CIDEr in medical report generation is missing.

5、	Since the IU-Xray dataset does not have an official training-test split, it is unclear if all the methods in comparison in Table 1 used the same training set on IU-Xray.

6、	The experimental comparison reported in Table 1 may not be fair, since the knowledge graph used in the proposed method seems to build upon an external dataset CheXpert, while other methods in comparison did not make use of this dataset (except PPKED[28] that uses CheXpert to finetune ResNet-152). At least the results under supervised setting are not comparable.In the literature, there are some medical report generation models that use the auxiliary CheXpert dataset, such as [R1], however, they were not included in the comparison.
[R1] Learning to Generate Clinically Coherent Chest X-Ray Reports, EMNLP2020



**Main Review:**

Originality: This paper proposes a knowledge graph based method for medical report generation and makes the first attempt to train the model in an unsupervised manner, which is certainly of value. However, there are also the following concerns/comments.  First, the authors argue that the paired image-report datasets are particularly labor-intensity and expensive to obtain, which is not very true. Indeed, the image-report pairs could be naturally generated in daily clinical routines when radiologists inspect the images and write the reports.  Although I agree with the importance of unsupervised learning, the unsupervised application scenario for medical report generation seems not as natural as general image captioning. It would be helpful if the authors could provide an example of unsupervised learning in clinic scenario.  Second, a significant claimed contribution is about the knowledge graph. As knowledge graph has been used in medical report generation, the authors need to clarify what their new contribution is in this aspect. This is not clear in the current paper as there is no experiment comparison with those knowledge-graph based medical report generation methods (such as [47, 22]) either, which seem to be close related works. Providing such information could help the reviewer better understands the originality of the paper.

Quality: This paper has two main technical contributions for medical report generation: unsupervised learning and knowledge graph generation. For the first contribution, it is interesting to see such an attempt for medical report generation, which should also be encouraged. For the second contribution, this motivation of incorporating knowledge graph is persuasive and the quantitative results reported in Table 1 are quite competitive (compared with those without using knowledge graph in supervised settings).  However, the comparison fairness needs to be clarified, since the knowledge graph seems to be built upon CheXpert dataset (14-class classicification), which is also used in the supervised manner. However, this external dataset is not used in the comparing methods (except [28]). In this way, I do not think the results in Table 1 (especially for supervised methods) are comparable. If I am wrong, please correct me.  The overall writing of this paper is good, but the clarify could be further improved (see Clarity).

Clarity: The clarity of the paper could be further improved with more explanations in motivation (such as an example clinical scenario for unsupervised report generation), contribution (such as the contribution in the knowledge graph part), how the unsupervised model could work (see Limitations), experimental settings (see Limitation), and the fairness of the comparison (see Limitations).

Significance: This paper studies an important task, medical image report generation, and tries to explore it in an unsupervised manner (unpaired images and reports), which is for the first time, and has referential value. The generated medical reports are regularized by the constructed knowledge graph that is expected to incorporate certain domain knowledge. The proposed model was also tested under semi-supervised and supervised settings although the fairness of the comparison needs further clarification (See Limitation).




**Time Spent Reviewing:**

4

---

> ### Author Response · Authors · 2021-08-10
> **Response to Reviewer@AGjp**
>
> Thanks for your helpful comments! If you have further concerns, please feel free to contact us.
>
> > **Q1**: The value of unsupervised medical report generation in clinic scenario.
>
> **A1**: In medical domain, the data are scattered in various hospitals, thus it's rather difficult to build a large dataset. As to the image-report pairs, the relatively large dataset MIMIC-CXR is not released until 2019. While other medical datasets like CheXpert only have images without reports. For the radiological reports given to patients and the case studies in textbooks, they only include the reports without the original images.
>
> > **Q2**: The new contribution of the knowledge graph.
>
> **A2**: Integrating knowledge graph in medical report generation has been explored in existing works [47][22]. However, their objectives and motivations of using knowledge graph are different from our work. Existing knowledge-graph based medical report generation methods aim to adopt the knowledge graph to boost the performance of supervised models. However, in our work, we aim to generate a medical report without using any coupled image-report training pairs, i.e., unsupervised medical report generation. A key challenge of unsupervised medical report generation is to bridge the gap between vision and language domains. To this end, we adopt the knowledge graph to create a latent space and propose a knowledge-driven encoder to project image and report to the same latent space (Line 50-55).
>
> > **Q3**: The comparison fairness needs to be clarified, since the knowledge graph seems to be built upon CheXpert dataset, which is also used in the supervised manner. The experimental comparison of our method with the existing knowledge-graph based medical report generation methods [47][22].
>
> **A3**: The knowledge graph in this paper is built upon the training set of the MIMIC-CXR dataset (Line 111), instead of the CheXpert dataset. Therefore, it's a fair comparison on the MIMIC-CXR dataset. The results in Table 1 and Table 2 show that our model can achieve state-of-the-art performances on the MIMIC-CXR dataset. For the other dataset IU X-ray used in our evaluation, for fair comparisons, we attempt to build our knowledge graph purely on the training set of the IU X-ray dataset. The following Table shows the comparison results of our methods, where the knowledge graph is purely built upon IU X-ray dataset (i.e., KGAE-Unsupervised (IU X-ray) and KGAE-Supervised (IU X-ray)), with the existing supervised methods, including the existing knowledge-graph based medical report generation methods [47][22], on IU X-ray dataset:
>
> | Methods | BLEU-1  | BLEU-2 | BLEU-3 | BLEU-4 | METEOR | ROUGE-L | CIDEr |
> | ---------- | :-----------:  | :-----------: | :-----------: | :-----------: | :-----------: | :-----------: | :-----------: |
> | NIC [38] | 0.216 | 0.124 | 0.087 | 0.066 | - | 0.306 | 0.294 |
> | AdaAtt [29] | 0.220 | 0.127 | 0.089 | 0.068 | - | 0.308 | 0.295 |
> | Att2in [33] | 0.224 | 0.129 | 0.089 | 0.068 | - | 0.308 | 0.297 |
> | Transformer [6] | 0.396 | 0.254 | 0.179 | 0.135 | 0.164 | 0.342 | - |
> | KERP [22] | 0.482 | 0.325 | 0.226 | 0.162 | - | 0.339 | 0.280 |
> | SentSAT+KG [47] | 0.441 | 0.291 | 0.203 | 0.147 | - | 0.367 | 0.304 |
> | R2Gen [6] | 0.470 | 0.304 | 0.219 | 0.165 | 0.187 | 0.371 | - |
> | PPKED [28] | 0.483 | 0.315 | 0.224 | 0.168 | 0.190 | **0.376** | 0.351 |\
> | |
> | KGAE-Unsupervised (IU X-ray) [Ours] | 0.425 | 0.271 | 0.185 | 0.114 | 0.123 | 0.310 | 0.301 |
> | KGAE-Supervised (IU X-ray) [Ours] | **0.519** | **0.331** | **0.235** | **0.174** | **0.191** | **0.376** | **0.387** |
> | |
>
> As we can see, by purely adopting the IU X-ray dataset to build our knowledge graph, our unsupervised model KGAE-Unsupervised can still outperform several supervised methods, i.e., NIC, AdaAtt, Att2in and Transformer, on most metrics, and our KGAE-Supervised can outperform existing knowledge-graph based medical report generation methods, i.e., KERP [22] and SentSAT+KG [47], and the state-of-the-art methods, i.e., PPKED [28] and R2Gen [6], across all metrics, including CIDEr.
>
> > **Q4**: The details of how to construct the knowledge graph.
>
> **A4**: In this paper, we build the knowledge graph with the following two steps: 1) first acquire a dictionary of key terms as nodes; 2) initialize the weights between terms according to the occurrence frequencies. These two steps don't need manual efforts. More details can be found in the supplementary material.
>
> > **Q5**:  Explain more why the unsupervised training method can work.
>
> **A5**: A key challenge of unsupervised medical report generation is to bridge the gap between vision and language domains without training on the pairs of images and reports. To this end, we adopt the knowledge graph to create a latent space and propose a knowledge-driven encoder, which includes a common mapping function, i.e., shared $\mathcal{F}$, to project images and reports to the same latent space (Line 50-55). As a result, our encoder can extract the image and report knowledge representations, i.e., the knowledge related to the image and report, they (image, report knowledge) share the common latent space, which allows our model to bridge the gap between vision and language domains without the training on the pairs of image and report.
>
> In implementations, as shown in Eq. (3) and Eq. (4), we acquire $\mathcal{G}_I$ and $\mathcal{G}_R$ by projecting $I$ and $R$ to the same latent space of the knowledge graph. By training $\mathcal{G}_I$ and $\mathcal{G}_R$ with the same classification task, we target to optimize the encoder to align both inputs $I$ and $R$. Here $\mathcal{F}$ in Eq. (4) plays a critical role. Under the unsupervised settings, the results of our KGAE-Unsupervised and KGAE-Unsupervised w/o $\mathcal{F}$ on the IU X-ray dataset are:
>
> | Methods | BLEU-1  | BLEU-2 | BLEU-3 | BLEU-4 | METEOR | ROUGE-L | CIDEr |
> | ---------- | :-----------:  | :-----------: | :-----------: | :-----------: | :-----------: | :-----------: | :-----------: |
> | KGAE-Unsupervised | **0.417** | **0.263** | **0.181** | **0.126** | **0.149** | **0.318** | **0.297** |
> | KGAE-Unsupervised w/o $\mathcal{F}$ | 0.265 | 0.151 | 0.105 | 0.074 | 0.097 | 0.268 | 0.162 |
> | |
>
> This shows that the shared $\mathcal{F}$ helps to bridge the gap between vision and language domains by encoding image and report knowledge representations into a common latent space. To further prove our arguments, in Figure 3, we adopt the t-SNE to visualize the image and report knowledge representations. We can find that there is a large gap between the raw image and report embeddings, while the encoded image and report knowledge representations better share the common latent space. The analysis proves that our proposed knowledge encoder can indeed encode images and reports to a common latent space (L309-312).
>
> > **Q6**:  The commonly used criteria CIDEr is missing.
>
> **A6**: In our submission, we follow [6] to report the performances of our methods which didn't include CIDEr results. Following your great advice, we further evaluate the CIDEr scores of our methods. In Table 1, the CIDEr scores of our KGAE-Unsupervised (i.e., KGAE (0%)) and KGAE-Supervised on IU X-ray/MIMIC-CXR are 0.297/0.118 and 0.370/0.255, respectively. The CIDEr scores of our methods still show that our unsupervised model KGAE-Unsupervised surpasses several supervised models (please see the Table in **Q3**), and our supervised model KGAE-Supervised surpasses the state-of-the-art models, e.g., PPKED (0.351/0.237 CIDEr scores on IU X-ray/MIMIC-CXR).
>
> > **Q7**: The IU X-ray dataset does not have an official training-test split.
>
> **A7**: Since the IU X-ray dataset does not have an official training-test split, it is important to ensure the comparison is fair on the IU X-ray dataset. To this end, most existing works attempt to randomly split the IU X-ray dataset into 70%-10%-20% training-validation-testing splits, and the splits may not be publicly available. In our work, we adopt the publicly available splits provided by R2Gen [6] and PPKED [28] for evaluation. Thus, it is fair to compare our method with R2Gen [6] and PPKED [28]. The results show that our model outperforms the state-of-the-art models R2Gen [6] and PPKED [28].
>
> > **Q8**: The comparison with the models that use the auxiliary CheXpert dataset, such as [R1].
>
> **A8**: Thank you for pointing out the comparison with the models that use the auxiliary CheXpert dataset, such as [R1]. In our work, like existing works [47][28], we only adopt the CheXpert dataset to fine-tune the CNN and not to build the knowledge graph. We also find that [R1][47][28] exploit the auxiliary CheXpert dataset, [16] exploits the auxiliary ChestX-ray14 dataset and [22][25] exploit the auxiliary ChestX-ray8 dataset. We will make corresponding revisions to cite and provide detailed comparisons of them.
>
> In detail, for the [R1] kindly provided by you, the experimental settings in [R1] are different from our paper. Specifically, [R1] randomly split the MIMIC-CXR dataset into 70%-10%-20% training-validation-testing splits, while our work follows the officially split. For other works, as shown in Table 1 of our paper and Table in **Q3**, our method outperforms them across all metrics, especially for [47] and [28], which also exploit the CheXpert dataset and incorporate the knowledge graph into the model.

---

> > ### Comment · Reviewer_AGjp · 2021-09-02
> > **---after reading the rebuttal---**
> >
> > The reviewer sincerely thank the authors for the clarification about the knowledge graph and especially their new experimental results on CIDEr. However, I would also like to point out that some of my major concerns are not removed after the rebuttal, detailed as follows.
> >
> > First, the motivation of unsupervised medical report is still not convincing. Although most public Chest X-ray datasets do not contain reports, the reports indeed existed in clinic for data collection and they were simply not shared as those datasets were not published for report generation tasks. For example, although Chest-Xray-14 dataset does not provide reports, its bounding box information was indeed mined from text reports,  A good motivation should come from the clinical demand, rather than public benchmarks. But as I mentioned, the unsupervised method may be useful for general image captioning, where image and text do not necessarily come together.
> >
> > Second, thank the authors for the clarification that ChestXpert was not used for knowledge graph construction. However, since it was used to fine-tune the CNN models, the fairness concern for the supervised results still remains, as most of the methods in comparison do not make use of ChestXpert. It is unclear how much this finetuning contributes to the final performance.
> >
> > Despite the above concerns, since this paper seems to be the first one exploring unsupervised medical report generation and it is in general technically sound, I am willing to increase my score from 5 to 6. It is suggested to add the results without ChestXpert finetuning in the final paper.

---

### Official Review · Reviewer_cvae · 2021-07-27

**Rating:** 7
**Confidence:** 4

**Summary:**

The authors propose an unsupervised method for the task of medical report generation. The model can be trained using unpaired images and reports. To bridge the image and language representations, the model uses a pre-constructed knowledge graph to create a latent space and their encoder could map images and reports to the same latent space.

Specifically, 1) in the knowledge graph, the nodes are N_{KG} (= 200) frequent clinical abnormalities and normalities. The edge weights are calculated by the normalized co-occurrence of different nodes computed from the MIMIC-CXR training set. The KG embedding is trained from a graph neural network. 2) The encoder is fine-tuned on the CheXpert dataset from ResNet-50 pre-trained on ImageNet. The encoder uses the attention mechanism where queries are image or report embeddings and keys/values are KG entity embeddings. The encoder is trained with multi-label classification with binary cross entropy loss for common radiographic observations classification. 3) The decoder is a transformer-based model augmented with a memory network. The decoder is trained from scratch by re-constructing a given report. During inference time, the encoder maps an image into the latent space, and the decoder generates a report from its latent representation.

The authors evaluated their method on two datasets: IU X-ray and MIMIC-CXR. The model achieved decent performance at the unsupervised setting. Furthermore, fine-tuned using 60% paired data (60% semi-supervised), the model reaches the previous SOTA performance. Fine-tuned using 100% paired data (supervised), the model sets the new SOTA with significant gains.


**Limitations And Societal Impact:**

See “cons” above.


**Main Review:**

**Motivation**: the paper proposes an unsupervised method for medical report generation, and for this task there may not be many image-report pairs existing. The task is similar to the unsupervised summarization task.

**Pros**:
- The paper proposes an interesting idea for unsupervised image captioning by using a knowledge graph to define a latent space which bridges image and language. This could be used in other unsupervised conditional text generation tasks.
- The paper shows extensive experiment results (overlapping metrics, clinical efficacy metrics, and human evaluation).
- The model achieves good performance even only trained with unpair data, and shows SOTA performance when it is trained with paired data.

**Cons**:
- This method needs to rely on a pre-designed knowledge graph. This may require re-designing the graph every time when the domain is changed.

**Typos**:
- Section 3.3 Semi-Supervised and Supervised Training Details: “where the formal can evaluate the performance of our approach” (formal → former)

**Some comments and questions for the authors**:
- How sensitive is the model to the predefined knowledge graph? Some quick experiments could be ablation studies of different KGs, i.e., KGs defined on IU X-Ray only, both MIMIC-CXR and IU X-Ray.
- One of the limitations is that the method needs to rely on a pre-designed knowledge graph. In the future study, if you are able to use a general medical KG and demonstrate that the model in this work consistently generalizes well to all tasks such as MRI and Dermoscopy with one KG, it clearly addresses the aforementioned limitation.
- Section 4.5 Qualitative Analysis, “Under the semi-supervised setting, the R2Gen … generates some repeated sentences of normalities”. In my humble opinion, the sentence repetition problem is because R2Gen is not pre-trained on a relatively large corpus. Therefore it is clearly under-trained when only fed by 20% training data. This is not a fair comparison to KGAE-Semi (20%) which is pre-trained from 225,528 reports, so there are no repeating sentences.
- Table 4, results with and without F are always compared without B, do you also have results to compare with F with B (10,000) and without F with B (10,000)?
- If I understand correctly, the models are separately trained for KG embedding, encoder, and decoder. Is it possible to combine them (or some of them) for an end-to-end training? Maybe it helps backpropagate information among KG embeddings, encoder, and decoder?


**Time Spent Reviewing:**

4 hrs

---

> ### Author Response · Authors · 2021-08-10
> **Response to Reviewer@cvae**
>
> Thanks for your helpful comments! If you have further concerns, please feel free to contact us.
>
> > **Q1**: How sensitive is the model to the predefined knowledge graph (KG)?
>
> **A1**: Thank you for pointing out a potential analysis point. We follow your constructive advice to evaluate the performances using different KGs defined on IU X-Ray only, MIMIC-CXR only, both MIMIC-CXR and IU X-Ray. Two tables below show the results of KGAE (0%) and KGAE-Supervised (100%) on the IU X-ray dataset:
>
> | Methods | KGs | BLEU-1  | BLEU-2 | BLEU-3 | BLEU-4 | METEOR | ROUGE-L |
> | ---------- | -----------  | :-----------: | :-----------: | :-----------: | :-----------: | :-----------: | :-----------: |
> | NIC [38] | - |0.216 | 0.124 | 0.087 | 0.066 | - | 0.306 |
> | AdaAtt [29] | - | 0.220 | 0.127 | 0.089 | 0.068 | - | 0.308 |
> | Att2in [33] | - | 0.224 | 0.129 | 0.089 | 0.068 | - | 0.308 |
> | |
> | KGAE (0%) [Ours] | IU X-Ray | **0.425** | **0.271** | **0.185** | 0.114 | 0.123 | 0.310 |
> | KGAE (0%) [Ours] | MIMIC-CXR  | 0.417 | 0.263 | 0.181 | **0.126** | **0.149** | 0.318 |
> | KGAE (0%) [Ours] | MIMIC-CXR + IU X-Ray | 0.419 | 0.260 | 0.180 | 0.124 | 0.143 | **0.321** |
> | |
>
> | Methods | KGs | BLEU-1  | BLEU-2 | BLEU-3 | BLEU-4 | METEOR | ROUGE-L |
> | ---------- | -----------  | :-----------: | :-----------: | :-----------: | :-----------: | :-----------: | :-----------: |
> | R2Gen [6] | - | 0.470 | 0.304 | 0.219 | 0.165 | 0.187 | 0.371 |
> | PPKED [28] | - | 0.483 | 0.315 | 0.224 | 0.168 | 0.190 | 0.376 |
> | |
> | KGAE-Supervised (100%) [Ours] | IU X-Ray | **0.519** | **0.331** | 0.235 | 0.174 | 0.191 | 0.376 |
> | KGAE-Supervised (100%) [Ours] | MIMIC-CXR | 0.512 | 0.327 | **0.240** | 0.179 | 0.195 | 0.383 |
> | KGAE-Supervised (100%) [Ours] | MIMIC-CXR + IU X-Ray | 0.505 | 0.323 | 0.239 | **0.181** | **0.198** | **0.385** |
> | |
>
> As we can see, our KGAE (0%) using different KGs can consistently outperform several existing supervised models, i.e., NIC, AdaAtt, Att2in, across all metrics. Similarly, our KGAE-Supervised (100%) with different KGs can also consistently outperform existing state-of-the-art models, i.e., R2Gen and PPKED.
>
> > **Q2**: Using a general medical KG to deal with all tasks such as MRI and Dermoscopy.
>
> **A2**: Thanks for the great suggestion. Since there are several public medical KG available, like DBPedia in the medical category (https://dbpedia.org/page/Category:Medical_specialties), it is a good direction to use one general KG for all tasks. We will work on it in future works. In particular, the good robustness of our proposed model to the knowledge graph (please see Tables in **Q1**) could enable our method to have the potential to be applied to other tasks such as MRI and Dermoscopy, where the image-report pairs could be much less or even unavailable.
>
> > **Q3**: Section 4.5 Qualitative Analysis.
>
> **A3**: Both R2Gen (20%) and our proposed KGAE-Semi (20%) adopt the same amount of image-report pairs (20%) for training. We agree with you that the superiority of the proposed KGAE-Semi (20%) may be due to its adoption of the additional independent reports for pre-training. Nevertheless, the results prove that our proposed approach can relax the dependency on the paired datasets and accept independent sets of images or reports in pre-training to boost the performance, while R2Gen can only accept coupled image-report pairs. Such advantages enable the application of our approach to the scenarios where the coupled images and reports pairs are scarce (MRI, Dermoscopy, Retinal, etc.).
>
> > **Q4**: In Table 4, the ablation results to compare with $\mathcal{F}$ with $B$ (10,000) and without $\mathcal{F}$ with $B$ (10,000).
>
> **A4**: Following your great advice, we conduct the ablation study on the IU X-ray dataset to compare Full Model (i.e., with $\mathcal{F}$ with $B$ (10,000)) and Full Model w/o $\mathcal{F}$ (i.e., without $\mathcal{F}$ with $B$ (10,000)). The results of Full Model and Full Model w/o $\mathcal{F}$ under both the unsupervised and supervised training settings are:
>
> | Methods | BLEU-1  | BLEU-2 | BLEU-3 | BLEU-4 | METEOR | ROUGE-L |
> | ---------- | :-----------:  | :-----------: | :-----------: | :-----------: | :-----------: | :-----------: |
> | KGAE (0%) | **0.417** | **0.263** | **0.181** | **0.126** | **0.149** | **0.318** |
> | w/o $\mathcal{F}$ | 0.265 | 0.151 | 0.105 | 0.074 | 0.097 | 0.268 |
> | |
> | KGAE-Supervised (100%) | **0.512** | **0.327** | 0.240 | 0.179 | **0.195** | **0.383** |
> | w/o $\mathcal{F}$ | 0.508 | 0.322 | **0.245** | **0.183** | 0.189 | 0.380 |
> | |
>
> As we can see, under the supervised setting (i.e., KGAE-Supervised (100%)), Full Model and Full Model w/o $\mathcal{F}$ achieve similar performances. However, under the unsupervised setting (i.e., KGAE (0%)), the Full Model w/o $\mathcal{F}$ receives significantly worse performance than the Full Model, which further proves the effectiveness of the shared $\mathcal{F}$ in mapping images and reports to the same latent space for unsupervised medical report generation.
>
> > **Q5**: End-to-end training.
>
> **A5**: In the supervised setting, we find that the end-to-end training on the image-report pairs indeed boosts the performance. However, under the unsupervised setting, our preliminary experiments show that the end-to-end training will significantly degrade the performance.
> We speculate that: in our proposed model, we separate the training of the KG embedding and the encoder from the training of the decoder, which targets to align the image and report to the same shared latent space under the unsupervised setting; While in the end-to-end training, the gradients from the decoder will cause the KG embedding and the encoder to bias towards the report domain, resulting in reduced performance.

---

### Author Response · Authors · 2021-08-10
**General Response to All Reviewers**

We sincerely thank all the reviewers (R) for their time, effort, and valuable comments. Overall, all reviewers agree that the idea of first attempting for unsupervised medical report generation is interesting and "is certainly of value" (R@AGjp). Both R@cvae, R@AGjp and R@S5sP agree that the overall writing of this paper is good and the motivation of our methods is persuasive (R@AGjp) and clear (R@S5sP). As for our experiments, R@cvae mentions that "the paper shows extensive experiment results", and both R@cvae, R@AGjp, R@S5sP agree that our methods not only achieve decent and competitive performance under the unsupervised setting, outperforming several supervised models on two benchmark datasets, but also set the new state-of-the-arts with significant gains under the supervised setting.

We respond to the remaining issues as follows. The questions will be addressed in our revision as replied but more concretely.
In summary, we sincerely thank all the reviewers for their valuable inputs.

---

### Decision · Program_Chairs · 2021-09-27

**Decision:**

Accept (Poster)

**Comment:**

Overview:
The paper proposes a novel model for generating medical reports using medical reports and medical images that are *not* paired. The coupling between the two domains is achieved via a shared latent space grounded in Knowledge Graph, a reasonable choice since medical reports incorporate substantial amount of prior knowledge from the medical domain. The authors demonstrate the utility of their model with extensive empirical results and improve performance over SOTA.

Review:
Overall, the reviewers appreciated the novelty of the unsupervised medical report generation model. The reviewers had minor concerns which were adequately clarified by the authors in their detailed and meticulous responses. A few illustrative examples include:
1) Concerns about sensitivity to domain on which the KG was trained. They provide detailed experimental results to show the impact.
2) Ablation study in Table 4 with higher B. The clarifications are consistent with the results in the paper.
3) Request for Cider metric. Provided.
4) Fairness of training KG on CheXpert. Addressed with KG trained w/o CheXpert.
5) Clarification of the motivation for using KG. Adequately addressed.
6) and so on.